# Effects of arousal and movement on secondary somatosensory and visual thalamus

Gordon H Petty[1,2,3†], Amanda K Kinnischtzke[1,2,3†], Y Kate Hong[1,2,3‡], Randy M Bruno[1,2,3]*

[1]Department of Neuroscience, Columbia University, New York, United States; [2]Kavli Institute for Brain Science, New York, United States; [3]Zuckerman Mind Brain Behavior Institute, New York, United States

**Abstract** Neocortical sensory areas have associated primary and secondary thalamic nuclei. While primary nuclei transmit sensory information to cortex, secondary nuclei remain poorly understood. We recorded juxtasomally from secondary somatosensory (POm) and visual (LP) nuclei of awake mice while tracking whisking and pupil size. POm activity correlated with whisking, but not precise whisker kinematics. This coarse movement modulation persisted after facial paralysis and thus was not due to sensory reafference. This phenomenon also continued during optogenetic silencing of somatosensory and motor cortex and after lesion of superior colliculus, ruling out a motor efference copy mechanism. Whisking and pupil dilation were strongly correlated, possibly reflecting arousal. Indeed LP, which is not part of the whisker system, tracked whisking equally well, further indicating that POm activity does not encode whisker movement per se. The semblance of movement-related activity is likely instead a global effect of arousal on both nuclei. We conclude that secondary thalamus monitors behavioral state, rather than movement, and may exist to alter cortical activity accordingly.

**\*For correspondence:**
randybruno@columbia.edu

†These authors contributed equally to this work

**Present address:** ‡Department of Biological Sciences and Carnegie Mellon Neuroscience Institute, Carnegie Mellon University, Pittsburgh, United States

**Competing interest:** The authors declare that no competing interests exist.

## Editor's evaluation

Sensory information reaches the neocortex through multiple anatomical pathways in the thalamus. Prior work has disagreed on whether these encode parallel components of the same sensory signals or differ in how they mix sensory signals with information about behavioral state. Studying the somatosensory system in awake mice, the authors provide evidence supporting the second view. The authors find similar state dependent activity in a higher order visual thalamic nucleus. This is a timely study in that many have observed state-dependent activity throughout the cortex and thalamus, but the mechanisms of this activity are incompletely understood. This study brings us closer to revealing the source of this signal by ruling out major excitatory inputs including afferents carrying movement information, feedback from the cortex and inputs from the colliculus in the midbrain.

## Introduction

Somatosensory, visual, auditory, and gustatory cortex are each reciprocally connected with a specific subset of thalamic nuclei. These nuclei can be subdivided into primary and secondary (often termed 'higher-order') nuclei (*Guillery and Sherman, 2002*; *Herkenham, 1980*; *Phillips et al., 2019*). The primary nuclei are the main source of sensory input to the cortex and respond robustly to sensory stimulation with low latency (*Chiaia et al., 1991*; *Constantinople and Bruno, 2013*; *Sherman and*

*Guillery, 2002*; *Wimmer et al., 2010*). Unlike primary nuclei, the secondary nuclei are interconnected with many cortical and subcortical regions, and their role in sensation and cognition is poorly understood.

In rodents, the facial whisker representation of primary somatosensory cortex (S1) is tightly integrated with two thalamic nuclei: the ventral posteromedial nucleus (VPM) and the posterior medial nucleus (POm) (*Deschênes et al., 2016*; *Petersen, 2007*). Compared to the primary nucleus VPM, the secondary nucleus POm has broader receptive fields, longer-latency sensory responses, and poorly encodes fine aspects of whisker touch such as contact timing and stimulus frequency (*Diamond et al., 1992*; *Masri et al., 2008*; *Moore, 2004*; *Moore et al., 2015*). It receives input from S1, motor cortex, posterior parietal cortex, the zona incerta, and many other subcortical regions in addition to brainstem afferents (*Chiaia et al., 1991*; *Olsen and Witter, 2016*; *Trageser and Keller, 2004*). Whereas VPM innervates cortical layer 4 and the border of layers 5B and 6, POm projects to the apical dendrites of layer 1 as well as layer 5A (*Wimmer et al., 2010*). POm is a stronger driver of layer 2/3 cells than cortico-cortical synapses and can enhance sensory responses in pyramidal neurons of layers 2/3 and 5 (*Mease et al., 2016*; *Zhang and Bruno, 2019*). POm is thus positioned to strongly influence sensory computations in S1 and do so in ways that are highly distinct from VPM. However, what POm activity encodes remains a mystery.

One possibility is that POm activity encodes self-generated movements, through either sensory reafference (stimulation of the sensory receptors by active movement) or motor efference copy (internal copies of motor commands), rather than extrinsic tactile sensations (*Yu et al., 2006*). If secondary thalamus were a monitor of movements (*Sherman and Guillery, 2002*), somatosensory cortex could use POm input to differentiate self-generated and externally generated sensory signals. However, recent studies in awake animals have observed that, in comparison to VPM, POm encodes whisker motion and contact less well (*Moore et al., 2015*; *Urbain et al., 2015*), which casts doubt on the hypothesis that secondary pathways exist to provide detailed motor information to cortex.

An alternative hypothesis is that secondary thalamic nuclei are key structures for monitoring behavioral state. For instance, several studies have noted that a subset of POm neurons is activated by pain (*Frangeul et al., 2014*; *Masri et al., 2009*), a powerful stimulus that can trigger a change in animals' state. Spatial attention is a more subtle form of behavioral state change and has been implicated repeatedly in studies of primate secondary visual thalamus (lateral pulvinar) (*Petersen et al., 1987*; *Saalmann et al., 2012*; *Wilke et al., 2010*). The rodent homolog to the pulvinar (lateral posterior nucleus, LP) is active during mismatch of movement and visual stimuli (*Roth et al., 2016*), which might reflect elevations in visual attention or even global arousal. These results raise the possibility that modulation by behavioral state is a general feature of all secondary nuclei.

Here, we investigate how afferent, corticothalamic, and collicular inputs—the three main excitatory pathways to secondary sensory thalamus—influence encoding of movements by POm in the awake mouse. We discovered that removing these circuits enhances rather than reduces modulation of POm activity by movements, suggesting that these pathways may mainly transmit signals of a nature other than movement. We further examine how POm activity relates to that of LP—which has not been directly compared before—to investigate general principles of secondary thalamus function. This comparison reveals that behavioral state, rather than movement itself, prominently dictates the activity of secondary thalamic nuclei.

## Results

We characterized the degree to which POm encodes whether or not an animal is whisking versus the fine details of whisking movements. We recorded juxtasomally from single neurons in head-fixed mice while acquiring high-speed video of the contralateral whisker field, from which whisker positions could be algorithmically extracted (*Figure 1A and B*; *Clack et al., 2012*). To measure slow aspects of whisking, we calculated whisking amplitude from the median angle across all whiskers. Whisking amplitude is defined as the difference in angle between the minimum and maximum protraction over the whisking cycle (*Hill et al., 2011*; *Moore et al., 2015*, see Methods). Whisking amplitude was used to determine periods of quiescence and whisking, as defined by periods of time when whisking amplitude exceeded 20% of the maximum for more than 250 ms (*Figure 1B*, shaded areas).

Whisking substantially elevated POm firing rates. We computed the mean firing rate for each cell during periods when the mouse was whisking versus quiescent (*Figure 1A* ; 22 POm neurons in five

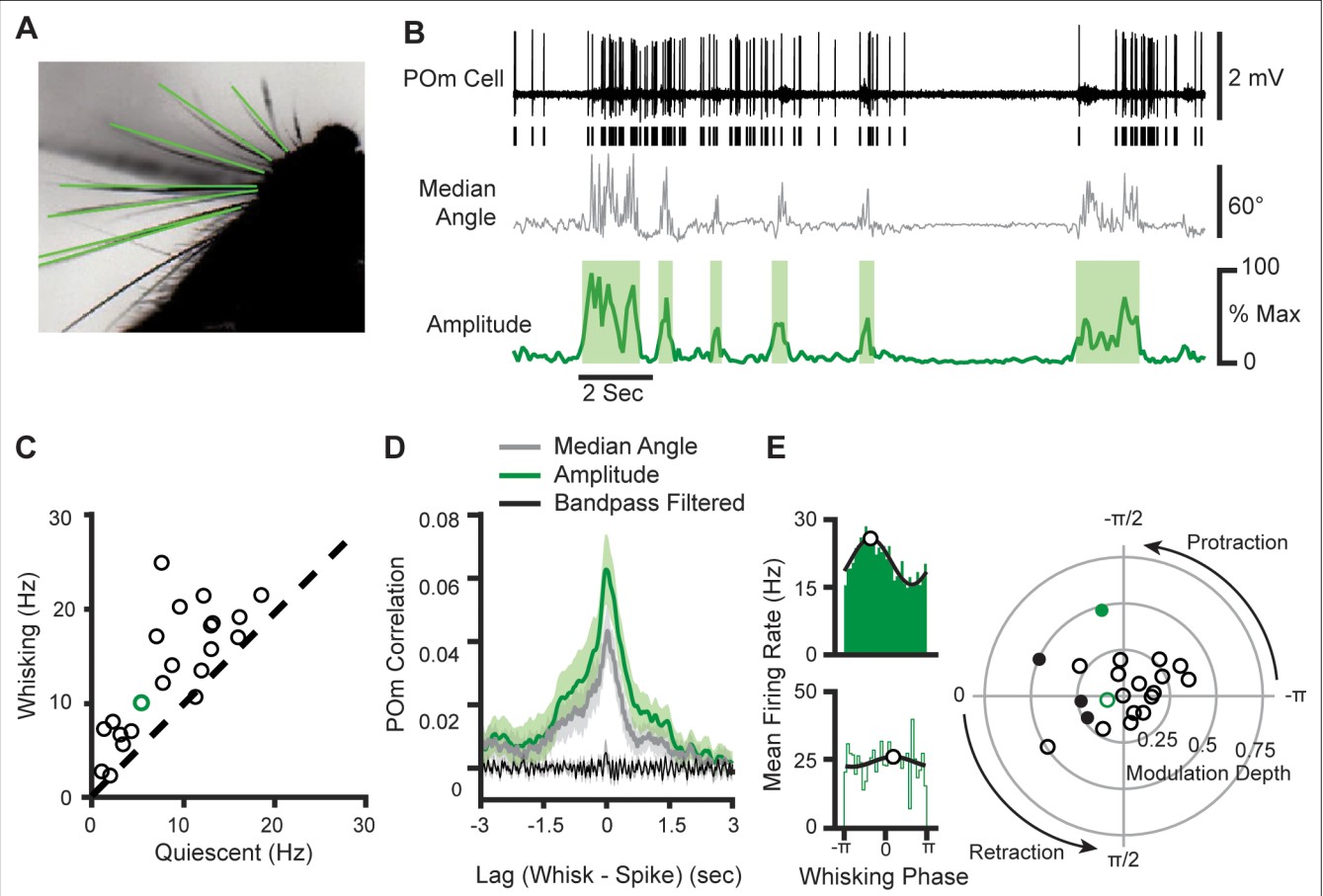

**Figure 1.** POm cells mainly track slow components of whisking activity. (**A**) An example frame from a video. Identified whiskers are highlighted in green. (**B**) Example traces of juxtasomal POm recording and whisking. The median angle of all whiskers in each video frame (middle, gray) was used to calculate the whisking amplitude (bottom, green). (**C**) Scatter plot of POm firing rates during whisking and quiescence (n=22 POm cells, five mice, increase from mean of 7.8 Hz to 12.4 Hz, or 58%, p<10⁻⁴, paired t-test). *Green*, cell in (**B**). (**D**) Cross-correlation of POm firing rate and whisking amplitude (green), angle (gray), and 4–30 Hz bandpass-filtered angle (black). Shading, SEM over cells. Cross-correlation is normalized such that autocorrelations at zero lag equal one. (**E**) *Left,* Firing rate as a function of phase in the whisking cycle for two example POm units. A sinusoid model (black) was fit to each cell to quantify preferred phase (white markers) and modulation depth. *Right,* A polar plot of modulation depth (radius) and preferred phase (angle) of each POm unit. *Filled circles*, cells with significant phase modulation (p<0.05, Kuiper test, Bonferroni corrected). Green circles correspond to the two examples.

The online version of this article includes the following figure supplement(s) for figure 1:

**Figure supplement 1.** Anatomical confirmation of recording locations.

**Figure supplement 2.** POm cells are coherent with whisking only at low frequencies.

mice; recording sites histologically confirmed post hoc, *Figure 1—figure supplement 1*). The firing rates of POm cells were significantly higher during bouts of whisking, increasing from a mean firing rate of 7.8 Hz to 12.4 Hz (58.5% increase, p<10⁻⁴, paired t-test). To understand which components of whisking might drive POm activity, we calculated the cross-correlation between POm firing rate and three features of whisking activity: the median angle across all whiskers (*Figure 1D*, gray), the amplitude metric which captures the slow envelope of whisking (green), and the median angle bandpass-filtered from 4 Hz to 30 Hz (black), which reflects fast protractions and retractions of the whiskers. We found that POm neurons had little correlation with the bandpass-filtered angle, but prominent correlations with both whisker angle and whisking amplitude around a time lag of zero. The strongest correlate of POm activity was whisking amplitude, suggesting that POm is coupled to the slow components of whisking, rather than individual whisk cycles.

To investigate the encoding of the fast components of whisking in POm, we analyzed whether individual cells preferred to discharge during a certain phase of the whisking bout. We quantified the phase of whisking by applying the Hilbert transform to the 4–30 Hz bandpass-filtered median

whisker. We identified the phase at which each action potential occurred during whisking and plotted distributions of firing rate as a function of phase. For each cell, we fit a sinusoid to characterize the cell's preferred phases (the phase of the whisk cycle that elicited the highest firing rate) and modulation depth (the degree to which phase impacts firing rate, measured as the peak-trough difference normalized by mean firing rate). *Figure 1E* shows the phase relationship of two example cells: one with significant coding (top) and the other insignificant (bottom). Most POm cells (18/22) resembled the non-phase coding example, having little or no modulation (right). Taken together, these results indicate that the majority of POm cells do not encode fast whisking dynamics such as whisker angle or the phase of the whisking cycle. Rather, they track overall whisking activity, that is, whisking versus quiescence.

To further characterize the relationship between whisking and POm activity, we analyzed the coherence of these signals in the frequency domain. A spectral density analysis of whisking angle indicated that the majority of whisking power is concentrated at low frequencies (<5 Hz) with a prominent peak at higher frequencies (8–13 Hz), corresponding to the dynamics of individual whisk cycles (*Figure 1—figure supplement 2A*). However, the mean-squared coherence of POm activity and whisking angle was significantly different from chance only in the low-frequency band (*Figure 1—figure supplement 2B*). For individual cells, mean coherence was significantly greater in the low-frequency band than in the high-frequency band (p=0.0004, paired t-test; *Figure 1—figure supplement 2C*). This was true even of cells with significant phase modulation depth. Thus, both time- and frequency-domain analyses confirm that POm output predominantly tracks the slow envelope of whisking.

One possible source of whisking-related activity is reafferent sensory input: when the mouse whisks, the self-generated movement could deform the whisker follicle and stimulate mechanoreceptors. To measure the degree to which POm activity is driven by the sensory reafference caused by whisking, we severed the buccal and upper marginal branches of the facial motor nerve on the right side of the face, contralateral to our recordings while taking video of the left (ipsilateral) side of the face (*Figure 2A*). This manipulation does not damage the sensory neurons and avoids the risk of inducing sensory neuron plasticity (*Shetty et al., 2003*). Mice were no longer able to move the right whisker pad but whisking on the left side of the face was unaffected (*Figure 2B and C*). Without whisker movement, there can be no reafferent sensory input from the right whisker pad. As in intact mice, firing rates of POm neurons in nerve cut animals were significantly higher during whisking bouts (*Figure 2D*, n=12, p=0.0007, paired t-test) and to a similar degree, increasing from quiescent firing rate of 11.6 Hz to a whisking rate of 16.7 Hz (44%). POm firing rates also correlated with ipsilateral whisking amplitude, at a similar magnitude and with a similar lag as in the contralateral whisker field in intact mice (*Figure 2E*). This demonstrates that the correlation of POm activity and overall whisking is not due to ascending reafferent information.

We also calculated the phase coding of the ipsilateral whisker field (*Figure 2F*) and compared it to the phase coding of the contralateral whisker field (*Figure 1E*). While average modulation depth was unchanged (*Figure 2E*, p=0.12, Wilcoxon rank-sum test), modulation depth by definition is bounded at zero, complicating analysis of medians close to zero. Indeed, there was a noticeable and statistically significant decrease in the range of modulation depths in the transected group (*Figure 2G*, p=0.0013, two-sample F-test), consistent with eliminating the largest modulation values. These results suggest that reafferent signals do not contribute to changes in POm activity reflecting the slow envelope of whisking (*Figure 2E*) but are responsible for the small population of POm cells carrying fast phase information (*Figure 2F*).

POm is reciprocally connected to several cortical areas, potentially making cortical output a strong driver of POm activity (*Chiaia et al., 1991*; *Diamond et al., 2008*; *Mease et al., 2016*). Primary motor cortex (M1) activity encodes the envelope of the whisking in absence of sensory input or motor output (*Ahrens and Kleinfeld, 2004*; *Fee et al., 1997*). Output from S1 and M1 could convey sensorimotor information, such as a motor efference copy, that would drive whisking-related activity, independent of ascending sensory input. To test the role of cortical output, we set out to study POm activity during optogenetic suppression of M1 (*Figure 3A*). We expressed halorhodopsin in all excitatory cortical neurons by crossing Emx1-Cre mice with a conditional halorhodopsin responder line, a technique we previously used to silence S1 (*Hong et al., 2018*). We first confirmed that the same technique silences M1 by recording it with a 64-channel silicon electrode array. Out of the 131 M1 cells recorded, 50 were fully silenced by the laser and a further 76 were substantially inhibited (*Figure 3B*, mean laser-on firing

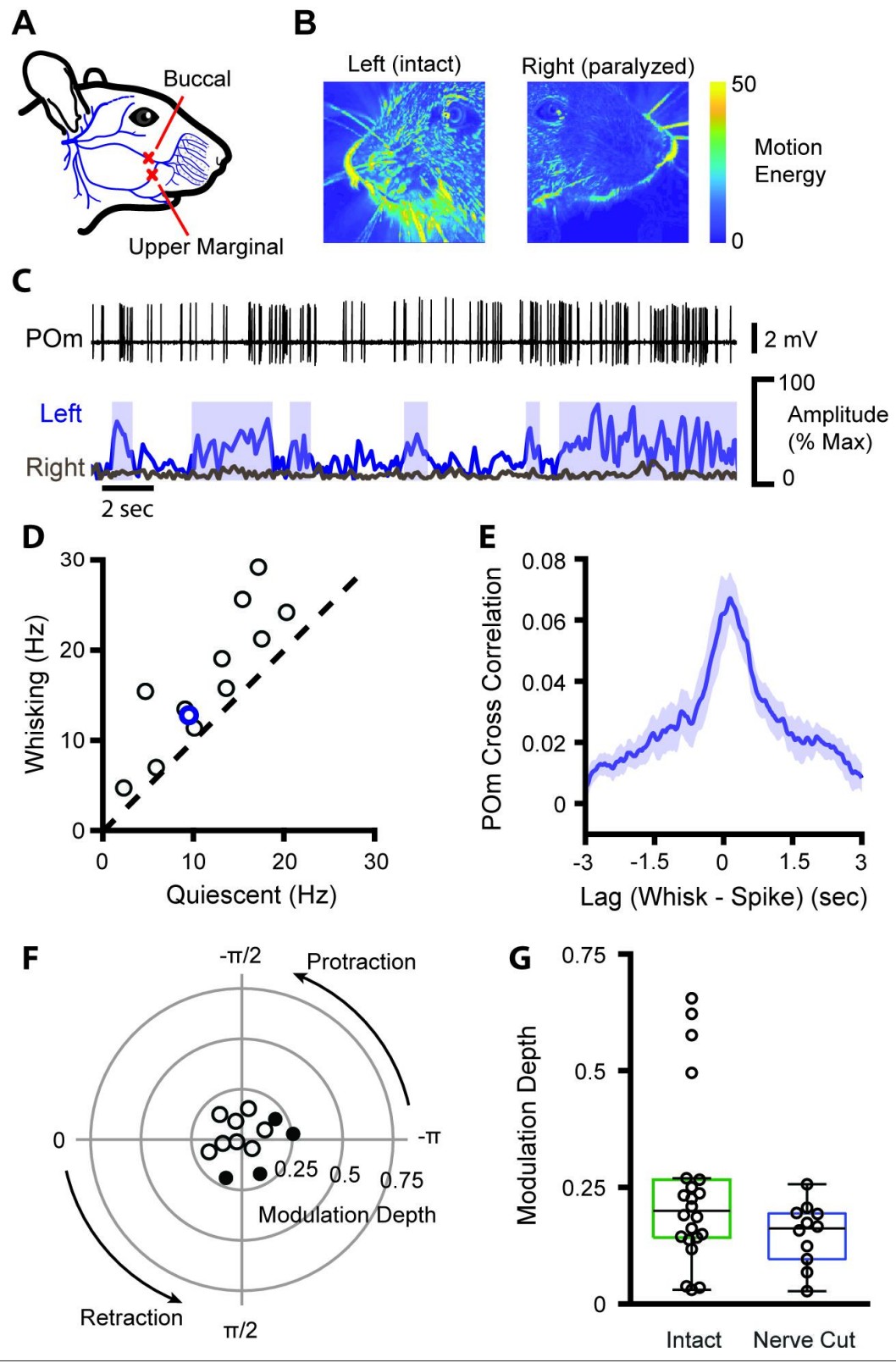

**Figure 2.** POm encodes whisking activity in absence of reafferent sensory input. (**A**) The buccal and upper marginal branches of the facial motor nerve were severed unilaterally, preventing whisker motion on the right side of the face. Adapted from *Heaton et al., 2014*. (**B**) Motion energy of the left (intact) and right (paralyzed) sides of the face, averaged over 3 min of video. Nerve cut greatly reduced the motion of both the whiskers and mystacial

*Figure 2 continued on next page*

*Figure 2 continued*

pad. (**C**) Example POm cell (top, black), ipsilateral (left side of face) whiskers (bottom, blue), and contralateral whiskers (bottom, gray). Blue boxes: periods of whisking as in *Figure 1B*. (**D**) Scatter plot of mean POm firing rate during whisking and quiescence. *Blue*, example cell in (**B**). Firing rates during whisking are significantly higher than quiescence (n=12 cells from two animals, quiescent mean: 11.6 Hz, whisking mean: 16.7 Hz, 44% change, p=0.0007, paired t-test). (**E**) Cross-correlation of POm firing rate and ipsilateral whisking amplitude. (**F**) Polar plot of modulation depth and preferred phase of each POm unit as in *Figure 1E*. *Filled circles*, cells with significant phase modulation (p<0.05, Kuiper test, Bonferroni corrected). (**G**) Modulation depth of POm cells in intact mice (green, as in *Figure 1E*) and after buccal nerve cut (blue). There was a significant difference in the variance of modulation depth between groups (p=0.0013, two-sample F-test).

rate=17.6% of baseline, median=1.04%). Inhibition was effective across cortical layers, and the laser decreased the firing rates of cells even deeper than 1200 μm from the surface (mean laser-on firing rate of deeper cells=60.1% of baseline, median=28.5%).

M1 suppression reduced the baseline firing rate of POm cells, but POm activity was still elevated during whisking regardless of whether the laser was on or off (*Figure 3C*). Surprisingly, suppressing M1 increased the correlation between POm firing rate and whisking amplitude (*Figure 3D*). This suggests that POm encoding of fine whisking kinematics arises from ascending sensory reafference rather than input from motor cortex.

To confirm that these effects were due to inhibition of M1 inputs to POm and not an artifact of optogenetic-induced changes in whisking behavior, we also recorded from cells in VPM. VPM, which does not receive direct projections from M1, was largely unaffected by M1 inhibition. We observed no effect of inhibition on VPM firing rates or cross-correlation between VPM activity and whisking (*Figure 3E and F*).

In a parallel set of experiments, we silenced S1 using the same cre-dependent halorhodopsin line (*Figure 4A*). This technique robustly blocks action potentials throughout all cortical layers of S1 in awake behaving mice (*Hong et al., 2018*). Silencing S1 reduced POm activity whether mice were whisking or quiescent (*Figure 4B*, n=11 cells, three mice; whisking p=0.0002, laser p=0.024, two-way repeated-measures ANOVA). As in the M1 experiments, the correlation between whisking amplitude and POm activity was, if anything, unchanged or increased by S1 silencing (*Figure 4C*, laser-off peak correlation=0.036, laser-on peak correlation=0.081). There was a tendency for S1 inhibition to reduce overall activity in VPM, possibly reflecting the known corticothalamic connections between S1 and VPM (*Bourassa and Deschênes, 1995*; *Hoogland et al., 1987*; *Killackey and Sherman, 2003*), but this effect did not reach significance (*Figure 4D*; p=0.1). Suppressing S1 had little impact on the correlation of VPM spiking and whisking (*Figure 4E*).

Thus, both optogenetic manipulations had qualitatively different effects on VPM and POm activity, consistent with the known anatomical differences in corticothalamic projections onto these two nuclei. Taken together, these results demonstrate that POm does not inherit information about whisking amplitude from M1 or S1. Rather, corticothalamic inputs appear to transmit signals other than whisker movements, and these additional signals normally reduce the correlation of POm activity with whisking amplitude and phase.

In addition to afferent inputs from brainstem and efferent inputs from cortex, POm receives excitatory projections from the superior colliculus (SC) (*Gharaei et al., 2019*), which could also provide a motor efference copy signal similar to known collicular circuits in the visual system (*Berman and Wurtz, 2010*). SC receives excitatory input from both the trigeminal brainstem and cortex and is whisking modulated (*Castro-Alamancos and Bezdudnaya, 2015*; *Castro-Alamancos and Favero, 2016*), making SC a potential source of whisking-related POm activity. To test this possibility, we performed bilateral electrolytic lesions in SC and subsequently recorded POm cells (*Figure 5A*). Whisking had similar effects on POm activity in both intact and lesioned animals (*Figure 5B*, n=49 cells from eight animals, 59% increase in mean firing rate, p<10$^{-9}$). POm firing rates of lesioned mice were overall higher than those of intact animals, independent of whether animals were whisking or quiescent (*Figure 5C*, lesion p<10$^{-3}$, whisking p<10$^{-5}$, two-way ANOVA). There was a slight tendency for SC-lesioned animals to whisk more frequently than intact animals, but this effect was not statistically significant (*Figure 5D*, p=0.35, Wilcoxon rank-sum test). While small parts of SC remained intact

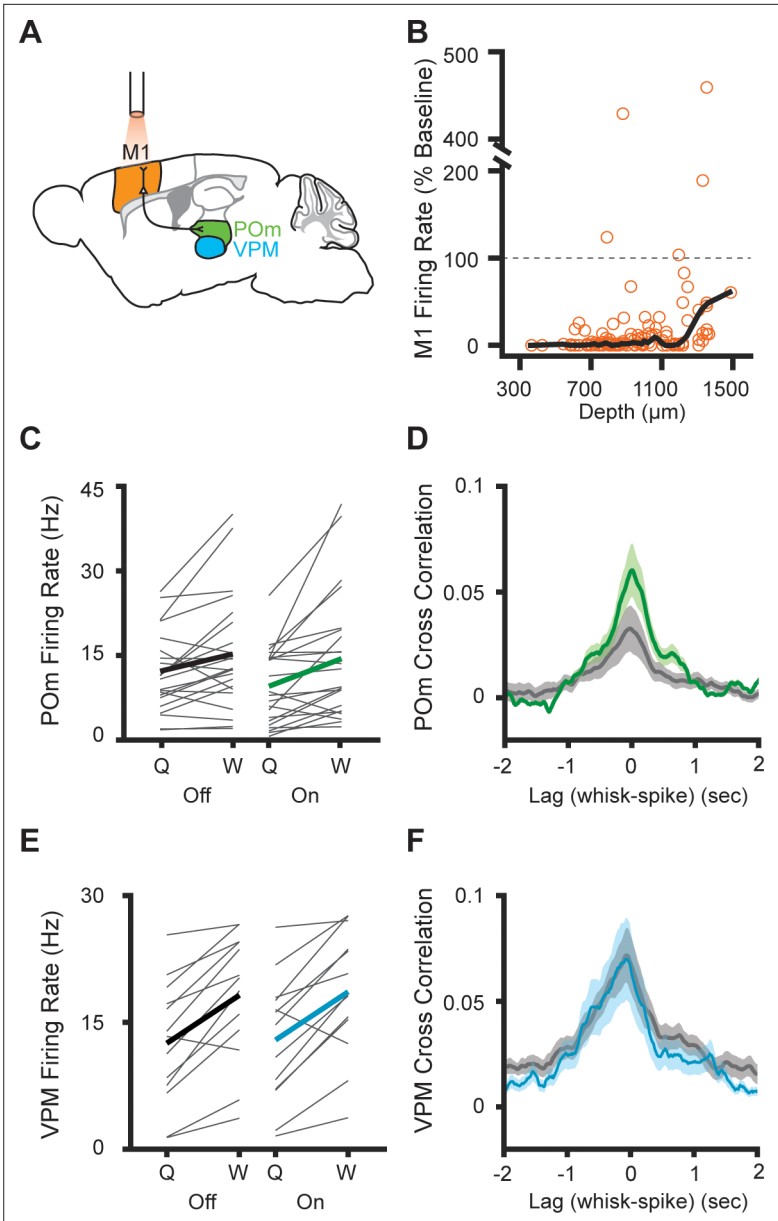

**Figure 3.** Inhibition of primary motor cortex increases POm correlation with whisking. (**A**) Experimental setup. M1 was optogenetically silenced while recordings were made from M1, POm, or VPM. Adapted from *The Mouse Brain Atlas in Stereotaxic Coordinates* (***Paxinos and Franklin, 2019***). (**B**) Effect of laser on M1 activity across cortical depth (n=131 cells, seven mice). The laser inhibited 96.2% of cells (mean laser-on firing rate 17.6%, median 1.04% of baseline). Black, lowess smoothed function. (**C**) Individual (gray) and mean (black or green) POm firing rates during whisking and quiescence when the laser is off or on (n=23 cells, three mice, whisking p=0.005, laser p=0.016, two-way repeated-measures ANOVA). (**D**) Cross-correlation of POm firing rate and whisking amplitude when the laser is off. The peak correlation was significantly higher when the laser was on (p=0.0018, paired t-test between peak values). (**E**) Individual (gray) and mean (black or blue) VPM firing rates during whisking and quiescence when the laser is off or on (n=13 cells, two mice, whisking p=0.0002, laser p=0.27, two-way repeated-measures ANOVA). (**F**) Cross-correlation of VPM firing rate and whisking amplitude (p=0.11, paired t-test between peak values).

(***Figure 5A***), these results with such large lesions suggest that SC is not responsible for the whisking-induced elevation of POm activity.

Neither reafference nor the most likely sources of motor efference copy explain the coarse modulation of POm by movement. This raises the question of whether POm encodes movement per se, or

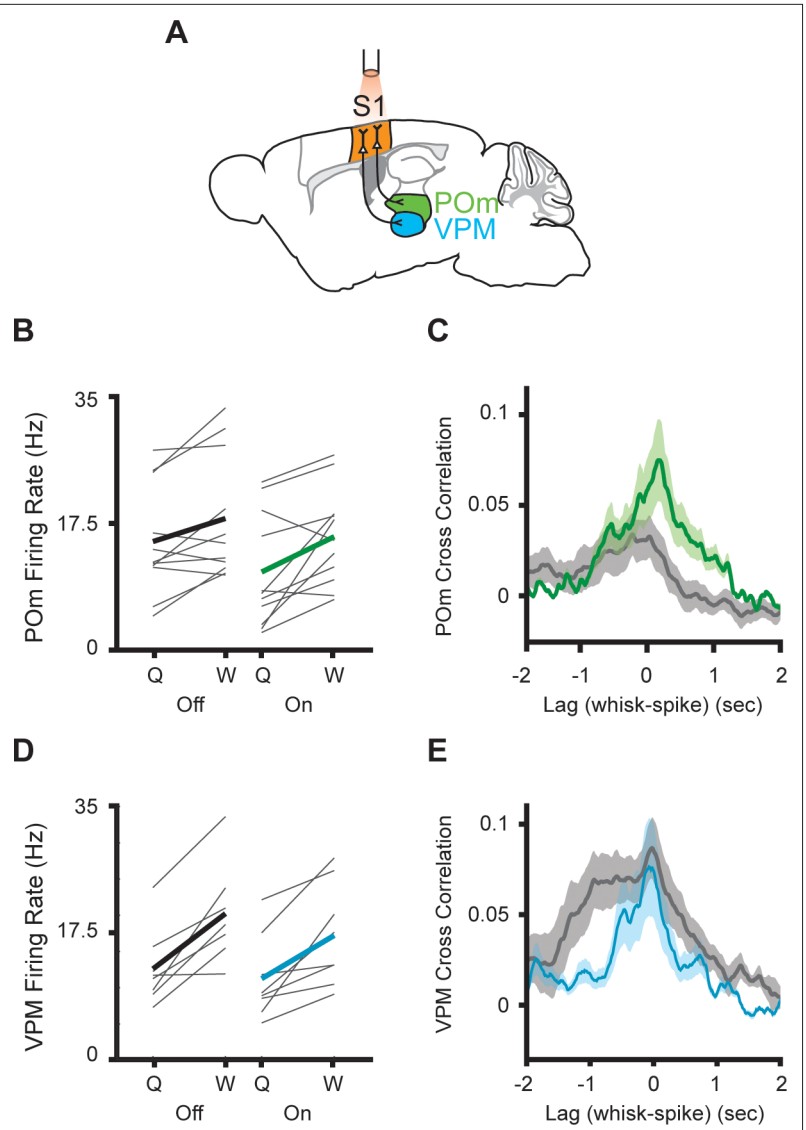

**Figure 4.** Inhibition of primary somatosensory cortex increases POm correlation with whisking. (**A**) Experimental setup. S1 was optogenetically silenced while recordings were made from POm or VPM. *The Mouse Brain Atlas in Stereotaxic Coordinates* (***Paxinos and Franklin, 2019***). (**B**) Individual (gray) and mean (black or green) POm firing rates during whisking and quiescence when the laser is off or on. (n=11 cells, three mice, whisking p=0.0005, laser p=0.03, two-way repeated-measures ANOVA). (**C**) Cross-correlation between POm firing rate and whisking amplitude when the laser is off (gray) or on (green) (p=0.15, paired t-test between peak values). (**D**) Mean VPM firing rate (n=8 cells, two mice, whisking p=0.001, laser p=0.11, two-way repeated-measures ANOVA). (**E**) Cross-correlation between VPM firing rate and whisking amplitude (p=0.48, paired t-test between peak values).

another variable that is coupled with whisking and other movements, such as arousal. To investigate this, we measured pupil diameter, which is a known metric of arousal. We acquired videos of the pupil and whiskers while recording from POm (***Figure 6A***). Pupil diameter was tightly correlated with whisking, with pupil dilation lagging whisking amplitude by 880 ms on average (***Figure 6B***). Pupil diameter also correlated with POm activity, to a similar degree as whisking and with a lag of 950 ms (***Figure 6C***, whisking amplitude peak correlation=0.052, pupil diameter peak correlation=0.071, p=0.23, paired t-test).

We reasoned that, if the modulation of POm by whisking was truly due to whisker movement rather than some other correlated variable, non-somatosensory thalamic nuclei would not be expected to track whisking. The secondary visual thalamic nucleus LP is the rodent homolog of the primate lateral

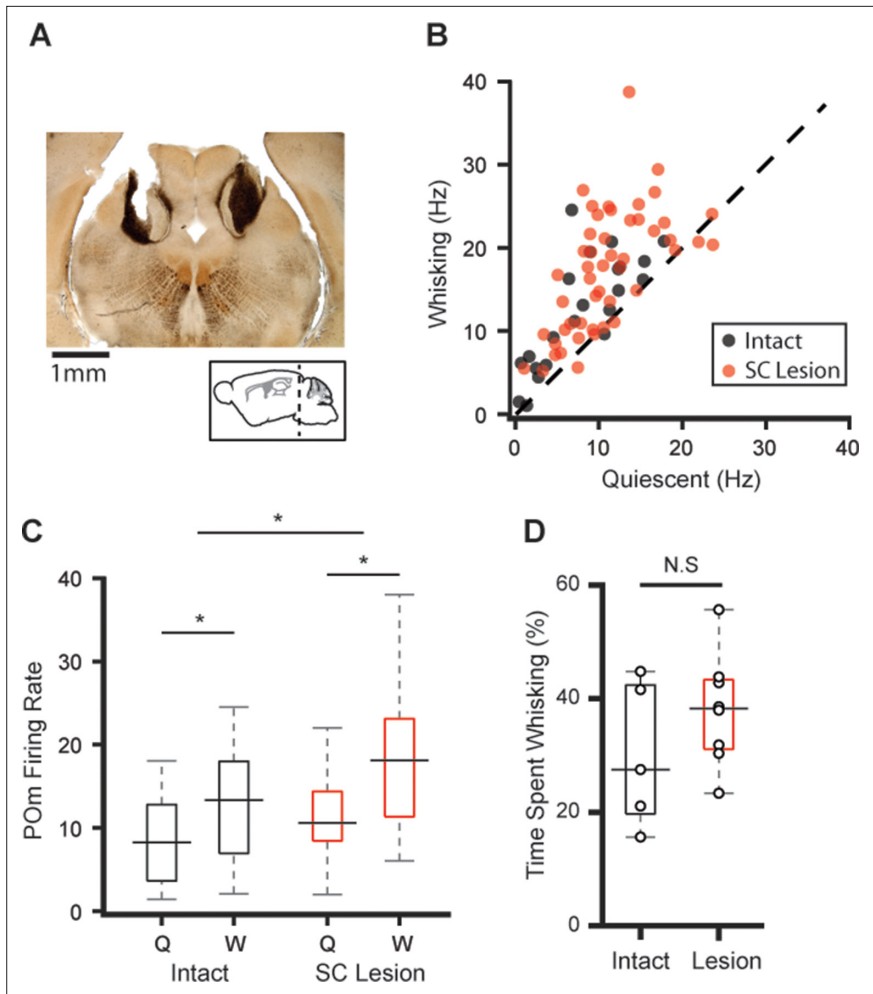

**Figure 5.** Lesions to superior colliculus (SC) do not reduce POm correlation with whisking. (**A**) Sample coronal section showing bilateral electrolytic lesion of SC. (**B**) Scatter plot of POm firing rates during whisking and quiescence in lesioned (red) and intact animals (black, data from *Figure 1*). Firing rates in lesioned animals were significantly higher during whisking (n=49 cells from eight animals, increase from mean of 10.9 Hz to 17.4 Hz, or 59%, $p<10^{-9}$, paired t-test). (**C**) Box plots of POm firing rates during whisking (W) and quiescence (Q) in intact (black) and lesioned animals (red). Pom firing rates in lesioned animals were higher than intact animals (whisking $p<10^{-5}$, lesion $p<10^{-3}$, two-way ANOVA). (**D**) Lesioned animals tended to spend slightly more time whisking, but this was not statistically significant (intact median=27.5%, n=5 mice; lesion median=38.3%, n=8 mice, p=0.35, Wilcoxon rank-sum test).

pulvinar. LP is primarily coupled with cortical and subcortical visual areas (*Nakamura et al., 2015*), rather than somatosensory ones. Because of their different connectivity, POm and LP are expected to carry separate sensorimotor signals related to somatosensation and vision, respectively. Therefore, LP would not be expected to encode whisker movement. By contrast, changes in behavioral state, such as overall animal arousal as suggested by our pupil measurements, might modulate all thalamic nuclei, including LP and POm.

We tested this idea by recording juxtasomally from LP neurons (*Figure 7A and B*; 29 cells from four mice). Surprisingly, we found that LP activity was significantly increased during whisking bouts (*Figure 7C*, increase from 13.0 Hz to 18.0 Hz, $p<10^{-4}$, paired t-test). Like POm, LP activity correlated with both whisking amplitude and median whisker angle with low latency (*Figure 7D*). Since changes in pupil diameter will cause more light to fall on the retina, the LP correlation with whisking might be an artifact of pupil dilation. To control for this, a subset of cells was recorded in low light. Under these darker conditions, the pupil was maximally dilated and did not change (*Figure 7B*), rendering input to the retina largely constant. However, these cells still showed an equivalent increase in firing

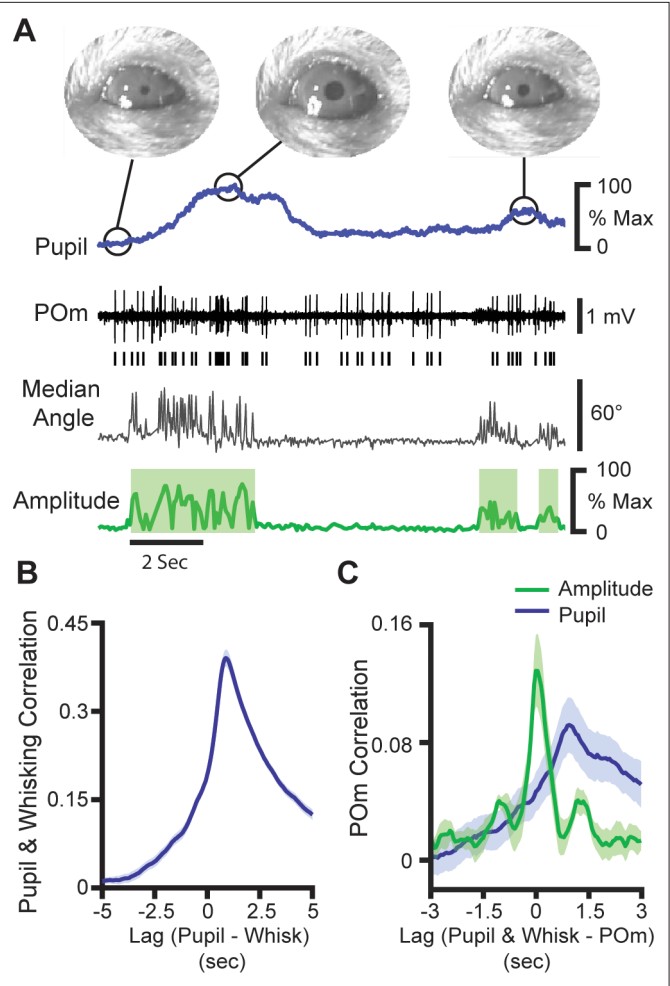

**Figure 6.** POm activity tracks pupil dynamics. (**A**) Sample recording of POm activity (middle, black) with concurrent ipsilateral pupil diameter (blue, top), median whisker angle (middle, gray), and whisking amplitude (green, bottom). (**B**) Cross-correlation of pupil diameter and whisking amplitude (30 recording sessions from seven animals). Errors bars are present but very small. (**C**) Cross-correlation of POm firing rate (n=10 cells from three animals) with whisking amplitude (*green*) and pupil diameter (*blue*).

rate during whisking (*Figure 7C*, orange; n=29 cells, four animals; increase from a mean of 13.0 Hz to 18.0 Hz, or 39%, $p<10^{-4}$, paired t-test). Thus, LP activity appears to track whisking independent of changes in visual input, which suggests that the effect in both nuclei is due to the arousal-whisking correlation rather than a direct effect of whisking.

The activity of individual LP cells was more coherent with the low-frequency components of whisking than high frequencies (*Figure 7—figure supplement 1A*). Overall LP-whisking coherence was identical to POm-whisking coherence in both the low- and high-frequency bands (*Figure 7—figure supplement 1B*).

Taken together, our results indicate that the slow component of whisking-related activity in POm is neither a consequence of ascending motion signals from reafferent mechanisms nor corticothalamic or colliculothalamic efferent mechanisms. We conclude instead that behavioral state, such as arousal may strongly dictate the activity of secondary thalamic nuclei, including POm and LP.

## Discussion

Our study tested the idea that the secondary somatosensory thalamus is a monitor of movements or motor commands and manipulated the multiple known pathways to POm that could mediate such signals. Juxtasomal recordings of POm cells revealed that this nucleus mainly tracks slow components

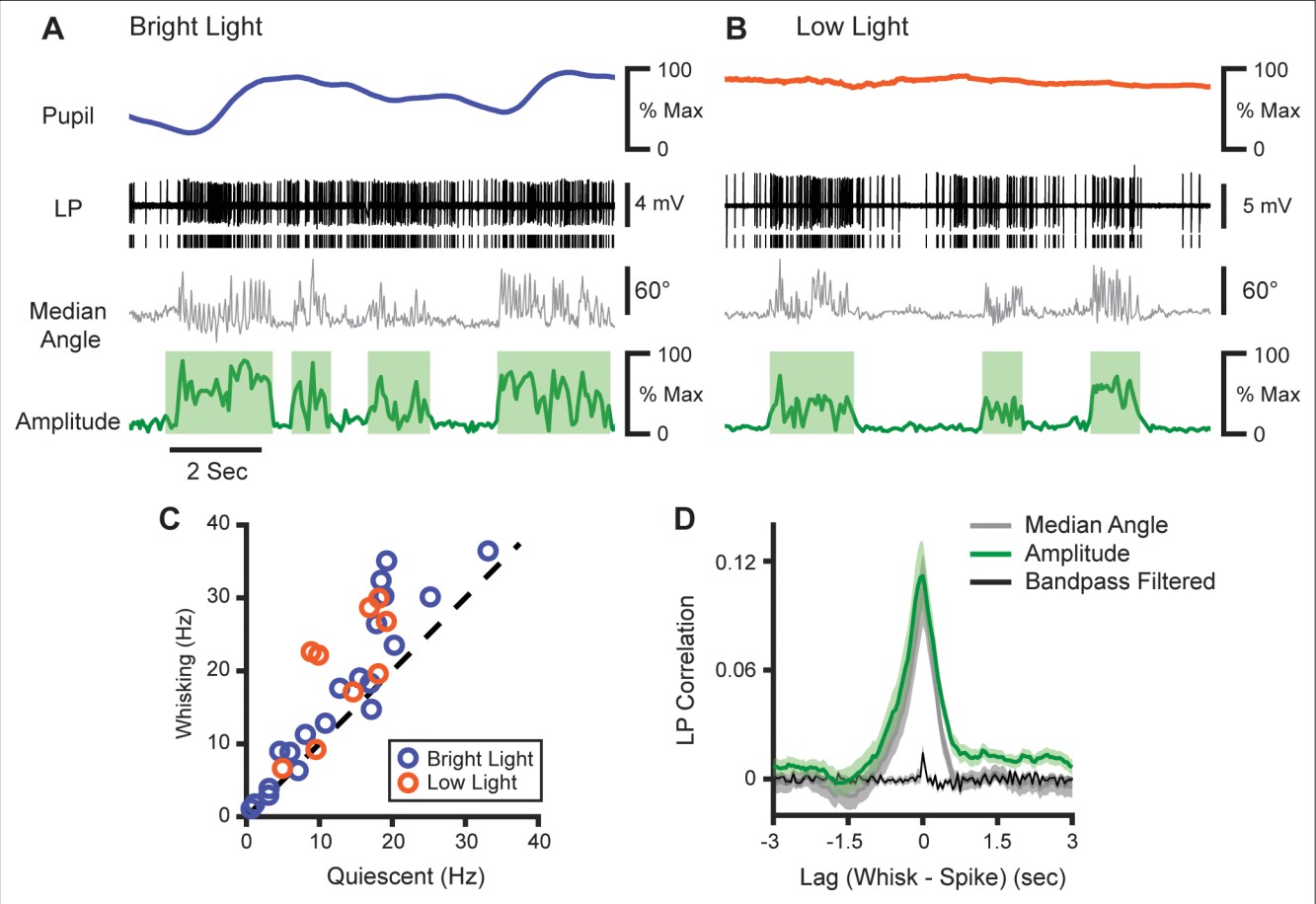

**Figure 7.** LP activity tracks slow whisker dynamics. (**A, B**) Sample recordings of two LP cells (black) recorded in normal light (**A**) or low light (**B**), with corresponding median whisker angle (gray) whisking amplitude (green), and pupil diameter (blue or orange). (**C**) Scatter plot of mean firing rate in LP cells during whisking and quiescence. *Blue*, cells recorded in bright light; *Orange*, cells recorded in low light (n=29 cells from four mice, p<10⁻⁴, paired t-test). (**D**) Cross-correlation of LP firing rate with whisking amplitude (green), median whisker angle (red), and 4–30 Hz bandpass filtered angle (black).

The online version of this article includes the following figure supplement(s) for figure 7:

**Figure supplement 1.** POm and LP cells have identical patterns of coherence with whisking.

of whisking, not detailed kinematics and that mouse POm firing rates are much higher during bouts of whisking than when a mouse is quiescent, consistent with other studies (***Moore et al., 2015***; ***Urbain et al., 2015***). POm activity mainly correlates with the slow change in whisking amplitude rather than the fast changes of the whisk cycle. Only a minority of our POm cells exhibited any whisking phase information, and phase encoding appeared to depend on sensory reafference. We have demonstrated that, by contrast, the overall elevation of POm activity by whisking is not due to sensory reafference from self-generated movements, as transection of the facial motor nerve did not uncouple POm activity from ipsilateral whisking. We showed that potential motor efference copy via corticothalamic pathways from S1 and M1 cannot account for whisking modulation of POm. Similarly, the phenomenon is independent of SC, the activity of which is linked to movement and orienting.

What appears to be movement-related activity in POm is likely instead a consequence of the encoding of behavioral state. Activity in secondary visual thalamus (LP) exhibits the same correlation with whisker movement that we observed in POm. Though it is possible that POm and LP separately encode correlated sensorimotor information, a more parsimonious explanation is that both POm and LP are modulated by arousal, which is naturally elevated during movement. Modulation of activity by the behavioral state may be a general property of all secondary thalamic nuclei. Future studies are needed to examine if this principle holds in auditory thalamic subnuclei and perhaps even thalamic nuclei connected to motor cortex and other frontal areas. Conceivably, some movement

correlations seen even in motor thalamus (*Guo et al., 2017*) may reflect various states more than specific movements.

The paralemniscal system has been speculated to be a parallel secondary afferent pathway (*Frangeul et al., 2014*; *Yu et al., 2006*). However, in anesthetized rats, POm does not appear to be sensitive to fine aspects of whisker touch, having very large receptive fields and long-latency responses (*Diamond et al., 1992*; *Trageser and Keller, 2004*). One might expect that very large synchronized movements of the whiskers, such as during whisking, would elicit a response from POm due to sensory reafference driving coarse receptive fields. However, paralyzing the face did not uncouple POm activity from ipsilateral whisking amplitude (*Figure 2*). Similarly, mouse barrel cortex is also modulated by whisking and quiescence in absence of sensory input: whisking is associated with a decrease in synchrony between layer 2/3 pyramidal cells in S1 and an increase in discharges by VPM, which is unaltered by bilateral transection of the infraorbital nerve sensory nerve (*Poulet and Petersen, 2008*). Manipulations of somatosensory thalamus strongly impacted cortical synchronization (*Poulet et al., 2012*). Further studies are needed to parcel out the extent to which thalamic contributions to cortical synchronization is due to inputs from VPM, POm, or both.

POm receives descending input from many cortical regions including M1 and S1. Conceivably these inputs could modulate ascending sensory input or provide the thalamus with a motor efference copy (*Mease et al., 2016*; *Sherman, 2016*). Similarly, LP and LGN axons in V1 exhibit eye movement-related signals (*Roth et al., 2016*). Previous studies in anesthetized rats have shown that cortical inactivation will silence POm, but not VPM (*Diamond et al., 1992*). Therefore, cortex might be the primary source of excitatory input to POm. However, we discovered that, in the awake mouse, silencing either M1 or S1 only slightly reduces the firing rate of POm cells and has little to no effect on VPM activity (*Figure 3*). We conclude that, while S1 and M1 provide significant excitatory inputs to POm, these inputs are not the sole drivers of POm activity during wakefulness.

Moreover, silencing these corticothalamic pathways increased rather than decreased the correlation between POm activity and whisking amplitude. If POm activity were primarily representative of a cortical efference copy, we would expect the opposite effect. While we cannot rule out the possibility that POm receives some efference copy from cortex, such input is not the cause of what at first appears to be whisking modulation. POm might instead be under equal or greater control of subcortical regions such as trigeminal brainstem complex, zona incerta, the thalamic reticular nucleus, and neuromodulatory brainstem centers—all of which receive inputs from broad areas of the nervous system (*Pinault and Deschênes, 1998*; *Trageser and Keller, 2004*).

As POm continues to track whisking in absence of both ascending sensory input and descending cortical input, we propose that the activity we observe is not sensorimotor in nature, but rather representative of thalamic coding of internal state. POm axons project to the apical dendrites of pyramidal cells (*Meyer et al., 2010*; *Wimmer et al., 2010*), where they might drive state-dependent changes in activity and synchrony. Arousal has dramatic effects on cortical dynamics (*Constantinople and Bruno, 2011*; *Reimer et al., 2014*; *Vinck et al., 2015*). We observed that pupil diameter, which closely tracks arousal, is highly correlated with whisking amplitude. Due to the coupling between pupil and whisking dynamics, they both correlate with POm firing rates (*Figure 6*). To dissociate the contributions of arousal and whisker movement, we took the novel approach of comparing POm dynamics with those of LP, the rodent homolog of the primate lateral pulvinar. We found a near-identical relationship between LP activity and whisking as we observed in POm (*Figure 7*), even though there is no known connectivity between LP and the whisker system. As for POm, these shifts in LP activity do not appear to be sensory dependent, as they persist even in low-light conditions where the pupil is maximally dilated and can no longer contribute to changes in retinal activity.

If state-dependent modulation of secondary thalamic nuclei is not derived from sensory reafference or motor efference copy from cortex or SC, the likely remaining candidates would include a large number of neuromodulators. While the neuromodulator acetylcholine is known to mediate the effects of whisking on activity in somatosensory cortex (*Eggermann et al., 2014*), our silencing experiments rule out any neuromodulation of primary somatosensory cortex as a mechanism. However, acetylcholine and norepinephrine also act directly on thalamic nuclei, such as VPM (*Aguilar and Castro-Alamancos, 2005*; *Hirata and Castro-Alamancos, 2010*). Zona incerta terminals within POm are regulated by acetylcholine (*Masri et al., 2006*) and are likely modulated in the same way within LP. Acetylcholine and norepinephrine both track pupil dynamics (*Reimer et al., 2016*) and are plausible

mechanisms of the POm-whisking correlation. In addition to these two well-studied modulators, many others are known to have direct effects on thalamus (*Varela, 2014*). Any of these could act directly on POm and LP or indirectly through ZI, TRN, and trigeminal brainstem nuclei, such as principalis and interpolaris. Future studies manipulating these many modulators and at different sites of action are needed.

The arousal effect we have described may be a more general version of modality-specific attentional effects that have been proposed for at least some secondary thalamic nuclei. In primates, pulvinar neurons respond strongest when stimuli are presented in attended regions of visual space (*Petersen et al., 1985*), and lesion of the pulvinar leads to deficits of selective attention during visual tasks (*Ward et al., 2002*; *Wilke et al., 2010*). Human patients with pulvinar damage exhibit spatial neglect, in which a stimulus can be perceived normally in isolation but is missed or distorted in the presence of neighboring stimuli (*Karnath et al., 2002*; *Snow et al., 2009*). By analogy, one might hypothesize that POm provides feedback that selects somatosensory stimuli for further cortical processing. Indeed, we and others have already demonstrated that activation of POm sensitizes cortical pyramidal neurons to the occurrence of subsequent tactile stimuli (*Mease et al., 2016*; *Zhang and Bruno, 2019*). Thus, POm affords control over the gain of the sensory responsiveness of somatosensory cortex circuitry. Selective enhancement of sensory responses by attention within a modality could be a general principle of all secondary thalamic functions.

Cortex-wide fluctuations in activity are known to correlate with various uninstructed movements (*Musall et al., 2019*). Cortical activity ceases in the absence of thalamic input (*Guo et al., 2017*; *Reinhold et al., 2015*), and secondary thalamic inputs to somatosensory cortex are stronger and longer lasting than corticocortical connections (*Zhang and Bruno, 2019*). Taking those studies and our study together suggests that secondary thalamus may be the underlying cause of the recently observed patterned fluctuations in activity across cortex. Our study directly tested the multiple known possible sources of afferent and efferent motor signals to secondary thalamus. None of these could explain apparent shifts in thalamic activity. Thus, behavioral state, rather than uninstructed movement, may be a primary driver of thalamic and cortical activity during movement.

Elevated firing rates in secondary thalamus due to arousal or attention could be useful for creating periods of heightened cortical plasticity. Recent studies have shown that repetitive sensory stimuli in anesthetized animals drive POm input to pyramidal neurons, which leads to enhancement of future sensory responses in cortex (*Gambino et al., 2014*). A potential mechanism of this is that disinhibition of apical dendritic spikes leads to long-term potentiation of local recurrent synapses among cortical pyramidal neurons (*Williams and Holtmaat, 2019*). Furthermore, an in vivo study found that associative learning can also potentiate long-range POm connections onto pyramidal neurons when subsequently measured in vitro (*Audette et al., 2019*).

It is conceivable that the arousal modulation of secondary thalamus that we have described is utilized by such processes. Our work opens avenues to examining potential links between arousal, attention, and plasticity.

## Materials and methods

All experiments complied with the NIH Guide for the Care and Use of Laboratory Animals and were approved by the Institutional Animal Care and Use Committee of Columbia University. Thirty C57BL/6 mice were used in these experiments.

## Surgery

Mice were anesthetized with isoflurane and placed in a stereotax. The skull was exposed, a thin layer of superglue was applied, and a custom-cut stainless steel headplate was attached using dental acrylic. A small (200 μm wide) opening was made on the mouse's left side at ~1.7 mm posterior to bregma and 1.4 mm lateral of the midline. A silver wire or screw was inserted over the frontal cortex of the same hemisphere as a ground electrode and covered with dental acrylic. The skin was sealed to the implant using superglue. Mice were allowed to recover from surgery for 5 days before habituation. Mice were habituated to the setup for 5 days by attaching their headplate to a holder on the recording table for 5–30 min each day, during which no recordings were performed.

## Electrophysiology

After habituation, a mouse would be recorded for 2–7 days. A glass micropipette (opening ~1.5 μm ID, shank ~60–80 μm OD over last 3–4 mm) was filled with artificial cerebrospinal fluid and inserted vertically into the brain using a micromanipulator. POm cells were typically recorded at microdrive depths of 2800–3600 μm relative to the pia, and LP cells were recorded at depths of 2100–2600 μm relative to pia. Recordings were made with a MultiClamp 700B amplifier (Molecular Devices), bessel filtered 300–10,000 Hz, and digitized at 16 kHz using custom Labview software (ntrode). At the end of some experiments, recording sites were labeled with a glass electrode coated in DiI inserted to a depth of 3600 μm relative to the pia.

Recordings of M1 were performed with 64-channel silicon electrode arrays (Cambridge NeuroTech H3 and H9). Arrays were inserted vertically to depths ranging from 1220 to 2000 μm from the cortical surface. Prior to recording, the tip of the array was dipped in DiI to label the recording site. Recordings were acquired with an OpenEphys acquisition system and software (*Siegle et al., 2017*), sampled at 30 kHz, and bandpass filtered from 2.5 Hz to 7.6 kHz.

## Videography

Whisker, pupil, and mystacial pad videos were made during electrophysiology and imaging using multiple PS3eye cameras running at 125 frames per second. Camera housings had been removed, and the lenses replaced with a 12 mm F2.0 lens (M12 Lenses Inc, part #PT-1220). Video was acquired using the CodeLaboratories PS3eye camera driver and the GUVCView software on a Linux computer.

## Optogenetics

Optogenetic silencing of cortex was performed using Emx1-Halo mice as previously described (*Hong et al., 2018*). Briefly, Emx1-IRES-Cre knock-in mice (Jackson Laboratories, stock #005628) were crossed to Rosa-lox-stop-lox (RSL)-eNpHR3.0/eYFP mice (Ai39, JAX, stock #006364), which express halorhodopsin after excision of a stop cassette by Cre recombinase. All mouse lines were maintained on a C57BL/6 background. Optogenetic experiments used mice that were heterozygous for the desired transgene as assessed by in-house genotyping. The locations of S1 and M1 were marked based on stereotaxic coordinates during headplate surgery, and the skull was thinned before recordings. Light was generated by a 593- or 594-nm laser (OEM or Coherent) coupled to a 200-μm diameter, 0.39 NA optic fiber (Thorlabs) via a fiberport, and the diamond-knife cut fiber tip was placed above M1 or S1.

## Nerve transection

The facial nerve was transected with the mouse under isoflurane anesthesia. A small (~5 mm) incision, centered ~5–8 mm ventral of the eye, was made in the skin. The buccal and upper marginal branches of the facial nerve were identified running from near the ear to the whisker pad, blunt dissected free of underlying tissue, and cut. The skin was closed with stitches and bupivacaine was applied. Mice were allowed at least 24 hr to recover from surgery before use in experiments.

## Superior colliculus lesion

The SC was lesioned bilaterally just prior to headplate implantation. Craniotomies were drilled over SC (0.5 mm anterior of lambda, 0.75 mm lateral of midline). A tungsten electrode (0.3–1.0 MΩ) was inserted to depths of 1 mm and 2 mm on each side, and 300 μA of current was delivered for 30 s at each lesion site. Mice were then implanted with a headplate and habituated as described above. Histology was used to confirm lesion size and location, and only recordings from mice with on-target lesions were analyzed.

## Histology

At the end of experiments, mice were deeply anesthetized with sodium pentobarbital and then perfused transcardially with 1× phosphate buffer followed by 4% paraformaldehyde. Brains were removed and sectioned on a vibratome into 100-μm-thick slices, or on a freezing microtome into 50-μm-thick slices. 100 μm slices were mounted directly on glass slides with mounting medium. 50 μm slices were stained in a solution of Cytochrome C (0.3 mg/ml), Catalase (0.4 mg/ml), and 3-3'-Diaminobenzidine (DAB, 0.583 mg/ml). Sections were incubated in this solution at 40°C for 30–45 min. Sections were washed five times in 1× phosphate buffer and mounted on glass slides with mounting medium.

## Data analysis

For juxtasomal recordings, putative action potentials were identified offline with custom MATLAB software. Spikes were then manually sorted with MClust (version 4.3). For silicon probe recordings, we used KiloSort3 to detect spikes and assign them to putative single units (*Pachitariu et al., 2016*). We then used Phy2 (https://github.com/cortex-lab/phy; *Rossant, 2021*) to manually inspect each unit. Units were assigned to the array channel on which its mean waveform had the highest standard deviation. Unit depth was assessed based on the microdrive depth of the probe tip from the cortical surface and the known distance of the channel from the tip. We identified putative inhibitory neurons based on the half-width of their mean waveform, that is, the time between maximum negativity and return to baseline (*Rodgers et al., 2020*). Units with a half-width below 0.25 ms were deemed narrow-spiking (putative inhibitory) and excluded from further analysis.

Whiskers were automatically tracked from videos using software (*Clack et al., 2012*). Custom MATLAB software was used to compute the median whisker angle. The median angle was bandpass filtered from 4 Hz to 30 Hz and passed through a Hilbert transform to calculate phase. We defined the upper and lower envelopes of the unfiltered median whisking angle as the points in the whisk cycle where phase equaled 0 (most protracted) or $\pm\pi$ (most retracted), respectively. Whisking amplitude was defined as the difference between these two envelopes. Periods of whisking and quiescence were defined as times where whisking amplitude exceeded 20% of maximum for at least 250 ms. Periods of time where amplitude exceeded this threshold for less than 250 ms were considered ambiguous and excluded from analysis of whisking versus quiescence. Sample sizes were chosen to be comparable to previous studies of POm (*Moore et al., 2015*; *Urbain et al., 2015*). The motion energy of the mystacial pad was defined as the mean absolute difference in pixel values between successive frames of a 3-min n video.

For cross-correlation analysis, whisking angle, amplitude, pupil, and spike vectors were binned with a 10-ms time bin. They were then normalized to have a mean of zero and standard deviation of one. Cross-correlations were again normalized such that the autocorrelation at a time lag of zero equaled one. To test the significance in changes between cross-correlation distributions (e.g., when comparing laser-off and laser-on conditions during cortical silencing), we found the lag of the peak correlation value for each distribution. We then performed paired t-tests between the correlation values of each cell at that time lag.

For each cell, each spike that occurred while the mouse was whisking was assigned a phase. The distribution of possible spike phases ($-\pi$ to $\pi$) was calculated using 32 equally sized bins. Using the same binning, we then calculated the distribution of phases observed in the video to determine the time the whiskers spent at various mean phases. We normalized the spike phase distribution by the phase distribution to calculate the firing rate as a function of phase. The modulation of the cell was characterized by fitting a sine function with a period of $2\pi$ to this rate function using least-squares regression. The modulation depth was calculated as the amplitude of the fitted sine wave divided by the cell's mean firing rate (*Moore et al., 2015*). To test the significance of this modulation, we compared the distributions of whisking phase and (unnormalized, unbinned) spike phase with a Kuiper test and a Bonferroni multiple-comparisons correction.

Whisking power spectra were computed using a multi-taper approach (Chronux, mtspectrumc, *Mitra and Bokil, 2008*). A power spectrum was calculated using the raw (unsmoothed, unfiltered) median whisker angle for each whisker video of the 11 mice used in *Figures 1, 2 and 7*. Mean power spectra were computed for individual mice. Whisking-spiking coherence for each POm and LP cell was similarly computed using a multi-tapered approach (coherencycpt). Shuffling was performed for each cell by offsetting the spike-time and whisking angle vectors by a randomly selected amount of time and computing the coherence between the offset vectors. Each cell was shuffled 50 times, and then all cells were averaged to obtain the mean shuffled coherence.

Pupil diameter was measured from video using custom MATLAB software. Videos were level-adjusted and thresholded to maximize the contrast between the pupil and the rest of the eye. The built-in imfindcircles() function was used to locate the pupil and measure diameter on each frame.

## Acknowledgements

The authors thank C.Kellendonk, A Das, C Rodgers, S Benezra and G Pierce for comments on the manuscript and D Baughman, B.C. Pil, L Jordan, and J Park for technical support. This work was supported by NIH R01 NS094659, R01 NS069679, and T32 EY013933.

## Additional information

### Funding

| Funder | Grant reference number | Author |
|---|---|---|
| National Institute of Neurological Disorders and Stroke | R01 NS094659 | Randy M Bruno |
| National Institute of Neurological Disorders and Stroke | R01 NS069679 | Randy M Bruno |
| National Eye Institute | T32 EY013933 | Gordon H Petty Amanda K Kinnischtzke |

The funders had no role in study design, data collection and interpretation, or the decision to submit the work for publication.

### Author contributions

Gordon H Petty, Conceptualization, Data curation, Formal analysis, Investigation, Writing - original draft, Writing - review and editing; Amanda K Kinnischtzke, Conceptualization, Formal analysis, Investigation, Methodology; Y Kate Hong, Methodology, Resources; Randy M Bruno, Conceptualization, Formal analysis, Funding acquisition, Supervision, Writing - original draft, Writing - review and editing

### Author ORCIDs

Randy M Bruno http://orcid.org/0000-0002-5122-4632

### Ethics

All experiments complied with the NIH Guide for the Care and Use of Laboratory Animals and were approved by the Institutional Animal Care and Use Committee of Columbia University (protocol AC-AAAY8462).

### Decision letter and Author response

Decision letter https://doi.org/10.7554/eLife.67611.sa1
Author response https://doi.org/10.7554/eLife.67611.sa2

## Additional files

### Supplementary files

• Transparent reporting form

### Data availability

Data have been deposited in Dryad.

The following dataset was generated:

| Author(s) | Year | Dataset title | Dataset URL | Database and Identifier |
|---|---|---|---|---|
| Petty G, Kinnischtzke A, Hong Y, Bruno RM | 2021 | Data from: Effects of arousal and movement on secondary somatosensory and visual thalamus | https://doi.org/10.5061/dryad.280gb5mr4 | Dryad Digital Repository, 10.5061/dryad.280gb5mr4 |

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
