## [Editor Report]

Sensory information reaches the neocortex through multiple anatomical pathways in the thalamus. Prior work has disagreed on whether these encode parallel components of the same sensory signals or differ in how they mix sensory signals with information about behavioral state. Studying the somatosensory system in awake mice, the authors provide evidence supporting the second view. The authors find similar state dependent activity in a higher order visual thalamic nucleus. This is a timely study in that many have observed state-dependent activity throughout the cortex and thalamus, but the mechanisms of this activity are incompletely understood. This study brings us closer to revealing the source of this signal by ruling out major excitatory inputs including afferents carrying movement information, feedback from the cortex and inputs from the colliculus in the midbrain.

---

## [Decision Letter]

**Decision letter after peer review:**

Thank you for submitting your article "Effects of arousal and movement on secondary somatosensory and visual thalamus" for consideration by *eLife*. Your article has been reviewed by 3 peer reviewers, and the evaluation has been overseen by Sacha Nelson as the Reviewing Editor and John Huguenard as the Senior Editor. The reviewers have opted to remain anonymous.

The reviewers have discussed their reviews with one another and with the Reviewing editor, and the Reviewing Editor has drafted this to help you prepare a revised submission.

Essential revisions:

1) As requested by reviewer #1, please provide some anatomical confirmation of the recording location. It would be helpful to provide a reply to reviewer #1's other anatomical comment about the movement of the mystacial pad.

2) Please address points 1,3,4,5 from reviewer 2. This should involve some additional analysis and textual changes.

*Reviewer #1 (Recommendations for the authors):*

(1) Please do the requested anatomy; it will improve the impact of the manuscript.

(2) For the "Main", the authors might consider recommending a review of whisking and the rodent whisker system, such as Petersen (Neuron 2007) and Kleinfeld and Deschenes (Neuron 2011).

*Reviewer #2 (Recommendations for the authors):*

1) How deep in the cortex did the inactivation occur? How much was the suppression? Referring to a previous paper is not sufficient for this important point.

2) The lesions of the superior colliculus were large, but incomplete. Would suction removal of the superior colliculus be better?

3) Scale bar for extracellular recordings is off for all figures. How can the action potentials be larger than they actually are (about 80-90 mV intracellularly)? Is this some type of consequence of the recording method used?

4) The authors should do a coherence analysis by frequency instead of a simple cross correlation. The relationship between whisking, activity, pupil diameter etc will be much higher at low frequencies, than high.

5) What are the possible other sources of the arousal/movement signal? Please discuss these in full.

*Reviewer #3 (Recommendations for the authors):*

I would encourage the authors to continue in this direction to define the role of POm versus other thalamic nuclei. Their study found that a group of cells in POm are sensitive to movement. It would be interesting to determine what is different between these "few" cells and the other insensitive POm cells. I would also encourage the authors to cite original relevant studies.

---

## [Author Response]

Essential revisions:1) As requested by reviewer #1, please provide some anatomical confirmation of the recording location. It would be helpful to provide a reply to reviewer #1's other anatomical comment about the movement of the mystacial pad.

We have addressed both anatomical issues. The revised manuscript contains new figures anatomically confirming recording locations in multiple structures (new Figure 1 —figure supplement 1). The revision also presents new data in Figure 2 documenting paralysis of the whisker pad after nerve cut.

2) Please address points 1,3,4,5 from reviewer 2. This should involve some additional analysis and textual changes.

In addition to the revisions of the text (points 3 and 5), we have performed new experiments documenting the extent of inactivation in motor cortex (point 1, Figure 3) and new analyses examining power spectra and coherence (new Figure 1 —figure supplement 2, Figure 7 —figure supplement 1).

Reviewer #1 (Recommendations for the authors):(1) Please do the requested anatomy; it will improve the impact of the manuscript.(2) For the "Main", the authors might consider recommending a review of whisking and the rodent whisker system, such as Petersen (Neuron 2007) and Kleinfeld and Deschenes (Neuron 2011).

Added.

Reviewer #2 (Recommendations for the authors):1) How deep in the cortex did the inactivation occur? How much was the suppression? Referring to a previous paper is not sufficient for this important point.

To address this important concern, we have added new experiments. We performed additional recordings of M1 in Emx1‐Halo mice to further characterize the silencing of M1 in this line. We used 64‐channel silicon probes spanning 1300 µm to assess silencing efficacy across cortical layers. Over 95% were significantly inhibited: Of the 131 M1 cells recorded in total, 50 were fully silenced by the laser and a further 76 were substantially inhibited (new Figure 3B, mean laser‐on firing rate = 17.6% of baseline, median = 1.04%). Inhibition was effective across cortical layers, and the laser decreased firing rates of cells even deeper than 1200 µm from the surface (mean laser‐on firing rate of deeper cells = 60.1% of baseline, median = 28.5%).

2) The lesions of the superior colliculus were large, but incomplete. Would suction removal of the superior colliculus be better?

Probably. However, the lesions are so large that it is hard to believe that POm‐projecting collicular neurons have their full complement of incoming synaptic connections, normal recurrent activity, and normal anatomical output to POm. At the same time, we observed no effect on POm‐whisking correlation. Colliculus is unlikely to be the underlying mechanism. Nevertheless, to address the reviewer’s concern, we specifically mention that lesions are large but incomplete and softened the conclusion.

3) Scale bar for extracellular recordings is off for all figures. How can the action potentials be larger than they actually are (about 80-90 mV intracellularly)? Is this some type of consequence of the recording method used?

We thank Reviewer 2 for pointing out this discrepancy. We have corrected the mistake (in Figures 1, 2, and 7).

4) The authors should do a coherence analysis by frequency instead of a simple cross correlation. The relationship between whisking, activity, pupil diameter etc will be much higher at low frequencies, than high.

We performed these new analyses (new Figure 1 —figure supplement 2, Figure 7 —figure supplement 1). As Reviewer 2 predicted, we see that POm and LP have significant coherence with whisking at lower frequencies (<5 Hz) and not higher frequencies (8‐13 Hz). To help nonexperts orient themselves to the data, we also included whisking power spectra.

5) What are the possible other sources of the arousal/movement signal? Please discuss these in full.

We have ruled out the likely glutamatergic sources of modulation. What we have not tested directly is direct neuromodulation of POm/LP and indirect neuromodulation of thalamic and brainstem inputs to these regions. We have expanded our Discussion along these lines and suggested future studies.

Reviewer #3 (Recommendations for the authors):I would encourage the authors to continue in this direction to define the role of POm versus other thalamic nuclei. Their study found that a group of cells in POm are sensitive to movement. It would be interesting to determine what is different between these "few" cells and the other insensitive POm cells. I would also encourage the authors to cite original relevant studies.

Our state dependent effect is larger than any truly kinematic effect we observed (such as phase dependence). We have revised the text to reinforce this.